# Oral Sheep Milk-Derived Exosome Therapeutics for *Cadmium*-Induced Inflammatory Bowel Disease

**DOI:** 10.3390/ijms26073299

**Published:** 2025-04-02

**Authors:** Zhimin Wu, Shuo Yan, Huimin Zhang, Zimeng Ma, Ruilin Du, Zhe Liu, Xihe Li, Guifang Cao, Yongli Song

**Affiliations:** 1Research Center for Animal Genetic Resources of Mongolia Plateau, College of Life Sciences, Inner Mongolia University, Hohhot 010020, China; wuzhimin_21@163.com (Z.W.); yanshuo202305@163.com (S.Y.); zhanghuimin202404@163.com (H.Z.); 15048197927@163.com (Z.M.); rallingd@163.com (R.D.); zheliu039@gmail.com (Z.L.); lixh@imu.edu.cn (X.L.); 2The State Key Laboratory of Reproductive Regulation and Breeding of Grassland Livestock, College of Life Sciences, Inner Mongolia University, Hohhot 010020, China; 3Inner Mongolia Saikexing Institute of Breeding and Reproductive Biotechnology in Domestic Animal, Hohhot 011517, China

**Keywords:** oral administration, sheep milk-derived exosome, cadmium, colitis, Hu Sheep

## Abstract

Cadmium (Cd) contamination in plants and soil poses significant risks to livestock, particularly sheep. Cd exposure often leads to severe gastrointestinal diseases in sheep that are difficult to treat. Milk-derived exosomes, particularly those from sheep milk (SM-Exo), have shown potential in treating gastrointestinal disorders, though their efficacy in Cd-induced colitis remains unclear. In this study, we investigated the therapeutic potential of SM-Exo in a Cd-induced colitis model. Hu sheep were exposed to Cd, and their fecal microbiota were collected to prepare bacterial solutions for fecal microbiota transplantation (FMT) in mice. The changes in gut microbiota and gene expression were analyzed through microbiome and transcriptomics. Our results showed that prior to treatment, harmful bacteria (e.g., *Bacteroides* and *Parabacteroides*) were increased in FMT mice. SM-Exo treatment increased beneficial bacteria, particularly *Lachnoclostridium*, and activated the Cyclic Adenosine Monophosphate (cAMP) pathway, upregulating genes like *Adcy1*, *Adcy3*, *CREB*, and *Sst*. These changes were linked to reduced Cd-induced cell death and alleviation of colonic inflammation. In conclusion, SM-Exo appears to be a promising treatment for Cd-induced colitis, likely through modulation of the gut microbiota and activation of the cAMP pathway.

## 1. Introduction

The rapid industrialization of modern society has increased environmental pollutants, which are closely linked to human habitats [1]. These pollutants directly affect human health and industries, with the livestock sector being especially vulnerable. Cadmium (Cd) is a common environmental contaminant, which is concerning due to its persistence in soil and plants [2]. It accumulates in the food chain, as plants absorb cadmium and animals ingest contaminated plants [3]. Beyond its risks in the food chain, cadmium is toxic to livestock, causing severe gastrointestinal disorders. Cadmium-induced colitis is difficult to treat and often results in prolonged illness. It impairs growth, reduces immunity, and negatively impacts animal welfare and farm productivity [4].

Oral administration is the most effective method for treating colitis in livestock [5]. Existing Nonsteroidal Anti-inflammatory Drugs (NSAIDs) for colitis are non-targeted, leading to systemic absorption and unwanted side effects, undermining their efficacy [6]. Biological therapies, particularly RNA-based drugs, are effective but expensive and face challenges in oral delivery due to degradation in the gastrointestinal tract and difficulty crossing the intestinal barrier [7]. Therefore, novel drug delivery technologies are needed to enhance the oral bioavailability of biologics.

Exosomes, small extracellular vesicles, play a crucial role in intercellular communication and the delivery of bioactive molecules [8]. Milk-derived exosomes have emerged as promising therapeutic agents for intestinal diseases due to their ability to withstand the gastrointestinal environment and facilitate targeted cellular uptake [9]. Notably, goat milk-derived exosomes have demonstrated therapeutic effects in mice colitis models, highlighting their potential as an oral treatment for colitis [10].

We established a cadmium-induced intestinal disease model in Hu sheep to explore the therapeutic potential of sheep milk exosomes. Previous research shows post-weaning Hu sheep are more susceptible to gastrointestinal diseases than pre-weaning sheep [11]. Despite advances in understanding the nutrients in sheep’s milk, many milk replacers and nutrient products are ineffective in treating cadmium-induced intestinal diseases. Therefore, we focused on exosomes, a unique component of sheep’s milk, as a potential therapy.

This study aims to isolate sheep milk exosomes from sheep and assess their therapeutic effects on cadmium-induced inflammatory responses in Hu sheep with the FMT mice model and Mouse Colonic Epithelial Cells (MCEC). Using high-throughput sequencing and metabolomic profiling, we analyzed gene and microbiome expression in mice with cadmium-induced colitis to explore the therapeutic potential of sheep milk exosomes and their impact on gut microbiota. This study offers insights into the role of sheep milk exosomes in modulating intestinal inflammation, paving the way for novel milk-based substitutes to improve intestinal health in livestock.

## 2. Results

### 2.1. Extraction, Purification, and Characterization of Sheep Milk-Derived Exosome

Exosomes were isolated and purified from sheep milk using a commercial kit. Exosomes were further characterized using transmission electron microscopy (TEM), nanoparticle tracking analysis (NTA), and Western blotting. TEM images (Figure 1A) revealed that sheep milk-derived exosomes (SM-Exo) exhibited typical round morphologies. NTA showed that the average particle size was 94.4 nm, with a concentration of 4.0 × 10^12^ particles/mL (Figure 1B). Western blotting confirmed the presence of exosomal markers CD63, CD81, and TSG101, and the absence of the exosome-negative marker calnexin (Figure 1C). These results confirm the successful isolation and purification of milk-derived exosomes.

### 2.2. Therapeutic Effect of Sheep Milk-Derived Exosomes in Fecal Microbiota Transplantation Mice

In a prior study, we established a cadmium-induced intestinal disease model in Hu sheep, which served as a donor for fecal microbiota transplantation (FMT) into mice (see Methods for details). We monitored the weight of mice in all three groups daily during the modeling process (Figure 2A). After 20 days, the median weight gain in the control (CON) group was 25.5%, significantly higher than in the cadmium (CD) group (−5.6%) and the exosome (EXO) group (10.7%). The median weight gain in the EXO group was significantly higher than in the CD group.

We also measured the spleen coefficients in the FMT mice. The results showed that the spleen coefficients in the EXO group were lower than in the CD group and similar to those in the CON group (Figure 2B). Additionally, colon length in both the CON and EXO groups was significantly greater than in the CD group (Figure 2C,G). Serum cholinesterase (CHE) levels were significantly lower in the CD group than in the CON and EXO groups (Figure 2D), while serum alanine aminotransferase (ALT) and cholesterol (CHOL) levels were significantly higher in the CD group compared to the CON and EXO groups (Figure 2E,F).

Hematoxylin and eosin (HE) staining of colonic tissues revealed loss of surface epithelium and goblet cells in the CD group. In contrast, the EXO group showed only mild loss of surface epithelium (Figure 2H). To further evaluate the therapeutic effects of sheep milk-derived exosomes, we performed Alcian Blue/Periodic Acid-Schiff (AB/PAS) staining. The results showed significantly increased purple staining of acidic mucus in the EXO group compared to the CD group, suggesting that SM-Exo treatment protected goblet cells’ mucus secretion from cadmium-induced damage (Figure 2I).

### 2.3. SM-Exo Treatment Alters the Intestinal Microbiota in FMT Mice

To investigate the effects of SM-Exo on the gut microbiota, we analyzed the fecal microbiota of FMT mice after oral administration of cadmium (CD) and SM-Exo. We collected cecal, proximal colon, and distal colon samples, which were then analyzed using 16S rRNA sequencing. The results showed significant differences in microbial diversity among the CD, control (CON), and SM-Exo (EXO) groups. Specifically, the richness index, along with indices such as Shannon, Chao1, and others, were utilized as metrics of species diversity to assess α diversity (Figure 3A). α diversity was significantly altered in the CD group compared to the CON and EXO groups (*p* < 0.05). β diversity analysis also revealed significant shifts in microbial composition after FMT, with SM-Exo administration inducing substantial changes in the gut microbiota profile within one week (*p* < 0.05) (Figure 3B).

We examined the microbial community in each group, identifying 1583 unique and shared microbial species (OTUs) in the CON group, 1614 in the CD group, and 1586 in the EXO group, with 1578 common microorganisms across all three groups. Additionally, the CON group has three OTUs, the Cd group has nine OTUs, and the EXO group has six OTUs (Figure 3C). At the phylum level, we observed an increase in *Bacteroidota* and *Verrucomicrobia* and a decrease in Firmicutes in the CD group. SM-Exo administration reversed these changes (Figure 3D). At the species level, *Lactobacillus* and *Akkermansia* were more abundant in the CD group, while *Bacteroides_thetaiotaomicron* decreased. SM-Exo treatment reversed these alterations. Besides, SM-Exo treatment significantly enriched *Lachnoclostridium*. (Figure 3D). Linear discriminant analysis (LDA) effect size (LEfSe) confirmed increased abundance of *Akkermansia*, *Lactobacillales*, and *Verrucomicrobiota* in the CD group, which was mitigated by SM-Exo treatment. Furthermore, SM-Exo administration increased the abundance of *Bacteroides* and *Lachnoclostridium* in the FMT mice (Figure 3E).

### 2.4. RNA-Seq Analysis Reveals the Mechanism of SM-Exo in Cd-Induced Intestinal Disease

Next, we performed RNA-seq analysis to examine gene expression changes in the colon of FMT mice treated with cadmium (CD) and SM-Exo. Principal component analysis (PCA) of the non-targeted metabolomics data revealed significant gene expression differences between the CD and CON groups, with SM-Exo reversing many of these changes (Figure 4A). In addition, we identified 509 differentially expressed genes (DEGs) specific to the CD vs. CON comparison, 24 DEGs specific to EXO vs. CON, and 527 DEGs unique to the EXO vs. CD comparison, with 3 DEGs common across all groups (Figure 4B). Volcano plots indicated that, compared to the CON group, 286 genes were upregulated and 1641 were downregulated in the CD group (Figure 4C). In the comparison between the CD and EXO groups, 1794 genes were upregulated and 206 were downregulated (Figure 4D).

To better understand the functions of these genes, we performed KEGG enrichment analysis. The results showed that genes involved in passive transmembrane transporter and channel activity were enriched between the CON and CD groups (Figure 4E), as well as between the CD and EXO groups (Figure 4F). Several key signaling pathways, particularly the cAMP signaling pathway, were found to respond to cadmium exposure and SM-Exo treatment. Genes involved in cAMP signaling, such as *Adcy1*, *Adcy3*, and *Sst*, were differentially expressed in the three groups (Figure 4G–I). The expression of *Adcy1*, *Adcy3* and *Sst* was significantly reduced in the CD group compared to the CON group, but SM-Exo treatment upregulated their expression.

In summary, RNA-seq analysis showed notable gene expression changes in cadmium-treated FMT mice, with SM-Exo treatment reversing many of these alterations, especially by regulating the cAMP signaling pathway.

### 2.5. SM-Exo Ameliorates CD-Induced Inflammation and Death in Cells

Before evaluating the anti-inflammatory effects of SM-Exo, we first confirmed its cellular uptake by MCEC cells in vitro. After 24 h treatment of MCEC cells with fluorescently labeled SM-Exo, confocal microscopy showed that the exosomes were absorbed by the cells, with fluorescence observed in the cytoplasm (Figure 5A). We next established a cadmium-induced inflammatory cell model by determining the optimal CD concentration (Figure 5B). The suitable rapidity of cell death was achieved at 0.1 µmol/L, and 0.1 µmol/L was chosen as the optimal CD concentration. CD treatment induced cellular death. However, SM-Exo treatment significantly reduced CD-induced cell death (Figure 5C), suggesting its anti-inflammatory effects and its ability to mitigate cell death in vitro.

## 3. Discussion

In this study, we employed a fecal microbiota transplantation (FMT) mouse model, using feces bacteria solution from Hu sheep exposed to cadmium, to explore the therapeutic effects of sheep milk-derived exosomes (SM-Exo) on cadmium-induced colitis. FMT mice exhibited typical colitis symptoms, such as significant weight loss, shortened colon length, inflammatory cell infiltration, and reduced mucus secretion, consistent with previous studies [12]. This model replicates the pathological features of intestinal diseases induced by environmental toxins like cadmium.

Oral treatments for colitis offer advantages, including ease of use and fewer side effects than systemic therapies [13]. However, challenges such as the impact of antibiotics on beneficial gut bacteria and food safety concerns limit the effectiveness of oral therapies. Additionally, biologic drugs struggle to maintain stability in the gastrointestinal tract. These challenges have driven interest in exosomes as a potential alternative for colitis treatment. Previous studies have demonstrated that porcine milk-derived exosomes can alleviate lipopolysaccharide (LPS)-induced intestinal damage [14], while goat milk exosomes offer protective effects in LPS-induced inflammation models [15]. This study shows that SM-Exo exhibits potent anti-inflammatory effects, providing a novel therapeutic approach for treating cadmium-induced intestinal inflammation.

To investigate the mechanisms behind these effects, we analyzed the intestinal microbiota in FMT mice using 16S rRNA sequencing. The results suggest that different treatment strategies significantly altered the microbial composition. At the phylum level, cadmium exposure reduced *Firmicutes* abundance, whereas SM-Exo administration restored this effect. At the species level, *Bacteroides* and *Parabacteroides* abundance decreased in the CD group, potentially contributing to colitis development. In contrast, *Lachnoclostridium* abundance was significantly higher in the EXO group, indicating a potential association with improved colitis symptoms. Previous studies have linked increased *Lachnoclostridium* abundance to reduced intestinal inflammation [16,17], supporting our findings that SM-Exo administration partially restores a healthy microbial profile.

The integration of RNA-seq and microbiome data provided valuable insights into the molecular mechanisms of cadmium-induced colitis. One key finding was the alteration of cAMP signaling, a pathway that regulates immune responses and intestinal homeostasis. In the CD group, cAMP pathway activity was significantly reduced, potentially contributing to immune dysregulation and inflammation. In contrast, SM-Exo treatment restored cAMP pathway activity, correlating with improved clinical outcomes. This finding aligns with previous studies showing that biologic therapies, such as anti-TNF agents, enhance cAMP signaling and promote mucosal healing in inflammatory bowel disease (IBD) patients [18]. The restoration of cAMP signaling by SM-Exo highlights the potential of targeting this pathway for therapeutic intervention.

Moreover, exosomes have emerged as important modulators of immune responses and intestinal barrier integrity in colitis [10]. This is also consistent with what we observed in vitro. SM-Exo increases uptake by MECE cells and alleviates the cell death rate induced by Cd SM-Exo is efficiently internalized by MECE cells and attenuates Cd-induced cell death. Previous research on mesenchymal stem cell-derived exosomes has shown their ability to reduce colonic inflammation by regulating macrophage polarization and enhancing intestinal barrier function [19]. While the exact mechanisms of exosome-mediated modulation of cAMP remain unclear, our findings suggest that oral SM-Exo administration upregulates key genes in the cAMP pathway, such as *Adcy1*, *Adcy3*, *CREB*, and *Sst*. This mechanism aligns with studies showing that exosomes from different sources can regulate cellular processes via the cAMP pathway. Additionally, exosomes from high-glucose-treated macrophages have been shown to activate inflammatory responses via the NF-κB pathway, which is closely linked to cAMP signaling [20].

While our study supports the therapeutic potential of SM-Exo in cadmium-induced colitis, certain limitations must be considered. First, we did not quantify bacterial-derived metabolites, including short-chain fatty acids (SCFAs) and bile acids, which are crucial markers of microbiota function. Future research will employ targeted metabolomics to elucidate their role in inflammation resolution. Second, although SM-Exo appears to alter gut microbiota composition, it remains unclear whether these changes result from direct bacterial modulation or secondary immune effects. Future studies will use co-culture models to determine whether SM-Exo directly promotes beneficial bacteria or alters microbial metabolite production. Additionally, although SM-Exo restored cAMP pathway activity, the exact mechanism of its regulation remains undefined. Future research should investigate whether SM-Exo modulates cAMP signaling through direct epithelial interactions or microbiota-derived metabolites. Furthermore, this study lacks direct comparisons between SM-Exo and conventional sheep colitis therapies, including antibiotics and biologics. Future studies should assess SM-Exo’s efficacy relative to conventional sheep colitis therapies and its potential as an adjunct treatment. Although this study was statistically powered, larger sample sizes and long-term studies are required to validate SM-Exo’s therapeutic potential. Evaluating SM-Exo in colitis models and assessing its safety will be essential for clinical translation.

In conclusion, this study highlights the complex interaction between host genetics, immune responses, and the gut microbiome in cadmium-induced colitis. The cAMP pathway is central to disease progression and response to therapy. Exosomes’ ability to modulate this pathway positions them as promising therapeutic agents for colitis treatment. Future research should focus on elucidating the mechanisms by which SM-Exo modulates the cAMP pathway and alleviates chronic inflammatory conditions like IBD.

## 4. Materials and Methods

### 4.1. Isolation of Exosomes from Sheep Milk

Exosomes were isolated from fresh sheep milk using a commercial kit (Umibio, UR52146, Shanghai, China). The milk was centrifuged at 5000× *g* a Sorvall ST 16R centrifuge (Thermo Fisher Scientific, Waltham, MA, USA) for 10 min to remove fat globules. The lower layer was collected and treated with corresponding reagents. The mixture was centrifuged at 5100× *g* and 25 °C for 20 min to remove casein. The supernatant was filtered through a 40 μm filter (Corning, Corning, NY, USA)and purified by ultracentrifugation at 35,000× *g* for 1 h, followed by 70,000× *g* for 3 h at 4 °C, using an Optima XPN-100 ultracentrifuge (Beckman Coulter, Brea, CA, USA). The middle layer was filtered through 0.8 μm, 0.45 μm, and 0.2 μm syringe filters (MilliporeSigma, Burlington, MA, USA). The exosome pellet was suspended in sterile phosphate-buffered saline (PBS) and stored at −20 °C until further use.

### 4.2. Identification of Exosomes

The exosomes were characterized using transmission electron microscopy (TEM) (HT7700, 80 kV, Hitachi, Tokyo, Japan) and nanoparticle tracking analysis (NTA). A 10 μL aliquot of the exosome suspension was stained with 2% phosphotungstic acid (Sigma-Aldrich, St. Louis, MO, USA) for 5 min at room temperature, and the morphology was examined by TEM. The particle size distribution was analyzed using Nano FCM (Flow NanoAnalyzer, Xiamen, China) as described by Mastoridis et al. [21].

### 4.3. Animal Model and Experimental Design

Eighteen two-month-old Hu sheep were selected from a commercial farm, housed individually in stainless steel metabolic cages, and allowed to acclimatize for 7 days. The animals were randomized into three groups and administered treatments by gavage over 21 days: the control group (CON, sterile saline) and the cadmium (Cd) group (2 g/kg). All sheep were euthanized 22 days after treatment initiation, and samples were collected immediately.

Four-week-old male C57BL/6JNifdc mice, pathogen-free and obtained from Weitong Lihua Laboratory Animal Science and Technology Co., Ltd. (Beijing, China), were housed under standard conditions with ad libitum access to food and water. Prior to fecal microbiota transplantation (FMT), the mice were treated with an antibiotic cocktail (vancomycin (0.5), ampicillin (0.5), metronidazole (0.5), and neomycin sulfate (1 g L^−1^; ABX Sigma-Aldrich, St. Louis, MO, USA)) for 7 days.

The FMT mice were divided into three groups: the normal group (fecal microbiota from healthy Hu sheep), the Cd group (fecal microbiota from cadmium-exposed Hu sheep), and the treatment group (50 mg/kg exosome solution with fecal microbiota). Mice were treated with 200 μL of fecal bacterial solution daily for 21 days.

### 4.4. Sample Size Determination and Power Analysis

To ensure adequate statistical power for detecting significant differences in inflammatory markers and microbiota composition, we conducted a power analysis using G*Power software (version 3.1, Heinrich-Heine-Universität Düsseldorf, Germany). Based on preliminary data and previous studies on exosome-based colitis treatments, we set an effect size (Cohen’s d) of 0.8, significance level (α) of 0.05, and statistical power (1 − β) of 0.9. The analysis indicated that a minimum of 8 animals per group was required to achieve sufficient power for detecting biologically meaningful differences. To account for potential variability and ensure robust statistical outcomes, we included *n* = 8–10 mice per group in the final experimental design.

### 4.5. MECE Cell Model Design

The MECE cell line was preserved by our lab. The cells were cultured in DMEM/F12 medium (Hyclone, Logan, UT, USA) supplemented with 10% fetal bovine serum (Biochannel, Nanjing, China) and 1% penicillin–streptomycin (Gibco, Waltham, MA, USA) at 37 °C in a 5% CO_2_ incubator. When the cell density reached 90%, the cells were digested and passaged with 0.25% trypsin (Gibco, Waltham, MA, USA). The MECE cell was treated with 0.1 μmol/L cadmium (Sigma-Aldrich, St. Louis, MO, USA) for 24 h to establish a Cd-induced cell model. To explore the effects of SM-Exo on the MECE death induced by cadmium, 200 μg/mL SM-Exo and 0.1 μmol/L cadmium were cocultured in 12-well plates during 24 h.

### 4.6. Cell Immunofluorescence Staining Protocol Using Calcein AM (Green) and PI (Red)

Cells were seeded onto sterilized glass coverslips in a 24-well plate and incubated at 37 °C with 5% CO_2_ overnight until reaching 70–80% confluence. After treatment under experimental conditions, cells were washed twice with pre-warmed PBS (37 °C, Gibco, Waltham, MA, USA) to remove residual medium. A working solution of Calcein AM (1 μM) (Abcam, Cambridge, UK, 148504) and Propidium Iodide (PI, 5 μg/mL) (Sangon, Shanghai, China A425259) was prepared in serum-free medium and added to the cells, followed by incubation at 37 °C for 15–30 min in the dark. After incubation, cells were gently washed with PBS to remove excess dye. Coverslips were mounted onto slides using an anti-fade mounting medium (Beyotime, Shanghai, China), and fluorescence imaging was performed using a fluorescence microscope (Leica SP8, Wetzlar, Germany). Calcein AM was detected at Ex/Em 488/515 nm, indicating live cells (green), while PI was detected at Ex/Em 535/617 nm, marking dead cells (red). Images were acquired and analyzed to assess live/dead cell populations. Throughout the procedure, minimal light exposure was maintained to preserve fluorescence intensity.

### 4.7. SM-Exo Staining with PKH67

Sheep milk-derived exosomes (SM-Exos) were labeled with PKH67 (Sigma-Aldrich, St. Louis, MO, USA) and purified by ultracentrifugation (120,000× *g*, 4 °C, 90 min, Beckman Coulter, Brea, CA, USA) to remove excess dye. Recipient cells were seeded on glass coverslips and incubated with PKH67-labeled M-Exos for 4–12 h. After washing with PBS, cells were fixed with 4% paraformaldehyde (Beyotime, Shanghai, China) and stained with DAPI (Beyotime, Shanghai, China). Exosome uptake was analyzed using a confocal microscope (Leica SP8, Wetzlar, Germany). A PBS control was included to account for background fluorescence.

### 4.8. Cell Culture

MECE Cells (ScienCell Research Laboratories, Carlsbad, CA, USA) were seeded onto sterilized glass coverslips in a 24-well plate and incubated at 37 °C with 5% CO_2_ overnight until reaching 70–80% confluence. After treatment under experimental conditions, cells were washed twice with pre-warmed PBS (37 °C) (Gibco, Thermo Fisher Scientific, Waltham, MA, USA) to remove residual medium. A working solution of Calcein AM (1 μM) (Invitrogen, Thermo Fisher Scientific, Waltham, MA, USA) and Propidium Iodide (PI, 5 μg/mL) (Sigma-Aldrich, St. Louis, MO, USA) was prepared in serum-free medium and added to the cells, followed by incubation at 37 °C for 15–30 min in the dark. After incubation, cells were gently washed with PBS to remove excess dye. Coverslips were mounted onto slides using an anti-fade mounting medium (ProLong Gold, Invitrogen, Thermo Fisher Scientific, Waltham, MA, USA), and fluorescence imaging was performed using a fluorescence microscope (Leica DMi8, Leica Microsystems, Wetzlar, Germany). Calcein AM was detected at Ex/Em 488/515 nm, indicating live cells (green), while PI was detected at Ex/Em 535/617 nm, marking dead cells (red). Images were acquired and analyzed using ImageJ software (v1.53t, National Institutes of Health, Bethesda, MD, USA) to assess live/dead cell populations. Throughout the procedure, minimal light exposure was maintained to preserve fluorescence intensity.

### 4.9. Organ Indices

The spleen and colon were collected and weighed, and the organ index was calculated using the Formula (1):Organ index (%) = (organ weight/body weight) × 100(1)

### 4.10. H&E and AB-PAS Staining

Colon tissue was fixed in 4% paraformaldehyde (Sigma-Aldrich, St. Louis, MO, USA), embedded in paraffin, sectioned, and stained with H&E for histological examination. Inflammation severity and mucosal damage were assessed by examining tissue slides at 100× magnification using a Leica DM2500 microscope (Leica Microsystems, Wetzlar, Germany).

For AB-PAS staining, colon sections were processed for deparaffinization and hydrated in a graded ethanol series. Sections were stained with Alixin blue (Sigma-Aldrich, St. Louis, MO, USA) and Schiff’s solution (Sigma-Aldrich, St. Louis, MO, USA) to assess mucus production and goblet cell presence, followed by hematoxylin counterstaining (Vector Laboratories, Newark, CA, USA). The stained sections were dehydrated, cleared with xylene (Sigma-Aldrich, St. Louis, MO, USA), and mounted with neutral gum (Solarbio, Beijing, China).

### 4.11. Preparation of Fecal Bacterial Solution for FMT

Feces from Hu sheep in the normal and diseased groups were collected and prepared as fecal microbiota solutions. The feces were mixed in sterile PBS (Gibco, Thermo Fisher Scientific, Waltham, MA, USA) (1 g/5 mL), filtered through double-layer sterile gauze, and centrifuged (2000 r/min, 5 min). The supernatant was discarded, and the pellet was resuspended in PBS to obtain the fecal bacterial solution. The solution was supplemented with 1 mL of sterile glycerol (Sigma-Aldrich, St. Louis, MO, USA) per 10 mL of bacterial suspension, aliquoted into centrifuge tubes, and stored at −80 °C.

### 4.12. Western Blotting

Total protein was extracted from SM-Exo using lysis buffer (Solarbio, Beijing, China, R0030) containing PMSF (Solarbio, Beijing, China, P0100) and phosphatase inhibitors (Thermo Scientific, Waltham, MA, USA, A32957). After centrifugation, protein concentrations were measured using a BCA assay (Thermo Scientific, Waltham, MA, USA, 23,227). Then, 10 µg proteins were separated by SDS-PAGE and transferred to nitrocellulose membranes (BIO-RAD, Hercules, CA, USA, 1620177). Membranes were blocked with 5% skim milk (BD, Franklin Lakes, NJ, USA, 232100) and incubated with primary antibodies (CD63 (25682-1-AP); TSG101 (28283-1-AP); Calnexin (10427-2-AP); and CD81 (27855-1-AP)) (ABclonal, Wuhan, China) diluted at a ratio of 1:2000 overnight at 4 °C. After incubation with HRP-conjugated secondary antibodies diluted at a ratio of 1:2000 (Signalway Antibody, College Park, MD, USA, L3012), the protein bands were detected using Pierce ECL Western blotting Substrate (Thermo Fisher Scientific, Waltham, MA, USA, 32209) and visualized with an E-BLOT contact imaging system (TouchImager, Suzhou, China, S2303063).

### 4.13. RNA-Seq Analysis

RNA-seq was performed on intact colon tissue samples. Total RNA was extracted using the TRIzol reagent (Invitrogen, Thermo Fisher Scientific, Waltham, MA, USA, 15596018) and purified with the RNeasy Mini Kit (Qiagen, Germantown, MD, USA, 74104). RNA integrity was assessed using an Agilent 2100 Bioanalyzer (Agilent Technologies, Santa Clara, CA, USA) with the RNA 6000 Nano Kit (Agilent Technologies, Santa Clara, CA, USA, 5067-1511), and only samples with RIN ≥ 7.0 were used for library construction. mRNA was enriched using the NEBNext Poly(A) mRNA Magnetic Isolation Module (NEB, Ipswich, MA, USA, E7490), followed by fragmentation at 94 °C for 5–7 min in the First Strand Synthesis Buffer (NEB, Ipswich, MA, USA, E7525). First-strand cDNA was synthesized using reverse transcriptase and random hexamer primers, and second-strand synthesis was performed using DNA Polymerase I and RNase H (NEB, Ipswich, MA, USA, E6111), followed by end repair and dA-tailing. The NEBNext Multiplex Oligos for Illumina primers (NEB, Ipswich, MA, USA, E7335, E7500) were ligated to cDNA fragments, and the final library was amplified by PCR and purified using AMPure XP beads (Beckman Coulter, Brea, CA, USA, A63881). Library quality was assessed using a Qubit dsDNA HS Assay Kit (Thermo Fisher, Waltham, MA, USA, Q32854) and an Agilent 2100 Bioanalyzer before sequencing on an Illumina NovaSeq 6000 platform (Illumina, San Diego, CA, USA), generating paired-end 150 bp (PE150) reads.

### 4.14. 16S rRNA Gene Sequencing of the Gut Microbiota

16S rRNA gene sequencing of fecal samples was performed by Wuhan Maiwei Metabolic Biotechnology Co., Ltd. (Wuhan China). The results were analyzed using the Maiwei Cloud platform (v1.0). Fecal DNA was extracted using the CTAB method and quantified by agarose gel electrophoresis. The V3-V4 region of the 16S rDNA gene was amplified using primers 341F and 806R. Libraries were constructed using the TruSeq^®^ DNA PCR Free Sample Preparation Kit (Illumina, San Diego, CA, USA) and sequenced on the NovaSeq 6000 platform.

### 4.15. Sequencing Data Processing

Raw sequencing data were filtered to obtain high-quality reads using fastp (v0.22.0, https://github.com/OpenGene/fastp, accessed on 1 February 2025) and FLASH (v1.2.11, http://ccb.jhu.edu/software/FLASH/, accessed on 1 February 2025). Clean tags were identified, and chimeric sequences were removed using vsearch (v2.22.1, Torbjørn Rognes, Oslo, Norway). The chimeric sequences were compared and identified with the species annotation database (https://github.com/torognes/vsearch, accessed on 11 February 2025).The final effective tags were used for downstream analysis.

### 4.16. Alpha Diversity Analysis

Alpha diversity indices, including Chao1, Shannon, Simpson, ACE, and PD-whole-tree, were calculated using the photoseq (v1.40.0) and vegan (v2.6.2) packages in R software (v4.2.0, R Foundation, Vienna, Austria). R was also used to generate dilution curves, rank abundance curves, and species accumulation curves. Intergroup differences in alpha diversity indices were assessed using both parametric and nonparametric tests. For comparisons between two groups, the *t*-test or Wilcoxon signed-rank test was applied. For more than two groups, Tukey’s test and the Kruskal–Wallis test were used.

### 4.17. Beta Diversity Analysis

Beta diversity was analyzed using the photoseq (v1.40.0) package in R (v4.2.0) to calculate UniFrac distances and construct a UPGMA clustering tree. Principal component analysis (PCA), principal coordinates analysis (PCoA), and non-metric multidimensional scaling (NMDS) were visualized using R software. PCA was conducted using the stats package (v4.2.0), while PCoA and NMDS analyses utilized the photoseq package (v1.40.0). Intergroup differences in beta diversity were evaluated using parametric and nonparametric tests. The *t*-test and Wilcoxon signed-rank test were used for comparisons between two groups, while Tukey’s test and the Kruskal–Wallis test were applied for multiple groups.

### 4.18. Differential Analysis

LEfSe analysis (v1.1.2, Boston, MA, USA) was performed with an LDA score threshold of 3.6 to identify significantly different taxa. Metastats analysis, conducted using Mothur software (Mothur software, v1.48.0, Ann Arbor, MI, USA), involved inter-group permutation tests at various taxonomic levels (phylum and species) to obtain *p*-values, which were corrected using the Benjamin-Hochberg false discovery rate (FDR) method to yield q-values. ANOSIM, MRPP, and Adonis tests were performed using the anosim, mrpp, and adonis functions in the R vegan package, respectively. AMOVA analysis was conducted using the amova function in Mothur software. Species with significant intergroup differences were analyzed using *t*-tests, and results were visualized using R software.

### 4.19. PCA

Unsupervised principal component analysis (PCA) was performed using the prcomp function in R (www.r-project.org, accessed on 11 February 2025). The data were preprocessed by applying unit variance scaling prior to PCA.

### 4.20. Statistical Analysis and False Discovery Rate (FDR) Correction

Benjamini–Hochberg FDR correction was applied to control for multiple comparisons in microbiome and metabolomic analyses. For 16S rRNA sequencing, α-diversity indices (Shannon, Chao1) were assessed using the Kruskal–Wallis test, and β-diversity (Bray–Curtis dissimilarity, weighted UniFrac) was analyzed via PERMANOVA (999 permutations) with R^2^ values reported. Differential taxa were identified using LEfSe (LDA > 3.0) with FDR-adjusted *p*-values.

For metabolomic analysis, one-way ANOVA followed by Benjamini–Hochberg post hoc correction was used. If normality assumptions were not met, Kruskal–Wallis tests with appropriate post hoc adjustments were performed. Statistical analyses were conducted in QIIME2 (v2023.2, Boulder, CO, USA), MetaboAnalyst (v5.0, Montreal, QC, Canada), and R (v4.2.0) using the vegan (v2.6.2) and phyloseq packages (v1.40.0).

Data are presented as mean ± SD. Group differences were tested using two-tailed Student’s *t*-test (with Levene’s test for variance equality) or one-way ANOVA. Welch’s *t*-test was applied for unequal variances. Statistical significance was set at *p* < 0.05, unless adjusted for multiple testing.

### 4.21. Ethics Statement

All animal procedures were approved by the Institutional Animal Care and Use Committee of Inner Mongolia University, China (approval number NMGDX (Wu) 2022-0003) and were performed in accordance with the National Research Council Guide for the Care and Use of Laboratory Animals.

## 5. Conclusions

Sheep milk-derived exosomes (SM-Exo) alleviate cadmium-induced colitis by modulating gut microbiota and may activating the cAMP pathway. SM-Exo improved colitis symptoms, enriched *Lachnoclostridium*, and upregulated cAMP-related genes, suggesting a role in immune modulation. These findings indicate SM-Exo as a promising candidate for sheep colitis treatment, warranting further studies to clarify its molecular mechanisms, long-term efficacy, and comparative therapeutic potential.

## Figures and Tables

**Figure 1 ijms-26-03299-f001:**
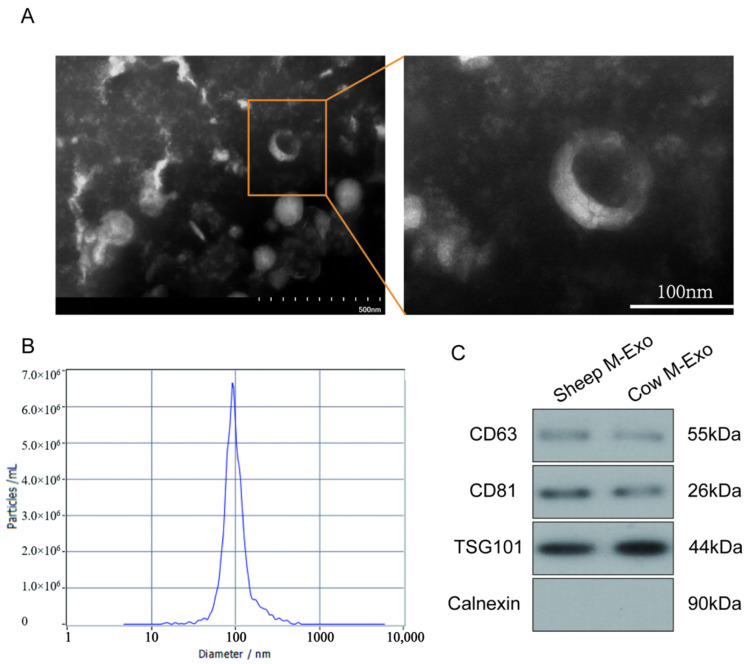
Characterization of sheep milk-derived exosomes. (**A**) TEM images of sheep milk-derived exosomes. Scale bars on (**left**) and (**right**) images represent 500 and 100 nm, respectively. (**B**) The size of the SM-Exo obtained was examined using an NTA. (**C**) Western blotting for the presence of Sheep M-Exo and Cow M-Exo markers.

**Figure 2 ijms-26-03299-f002:**
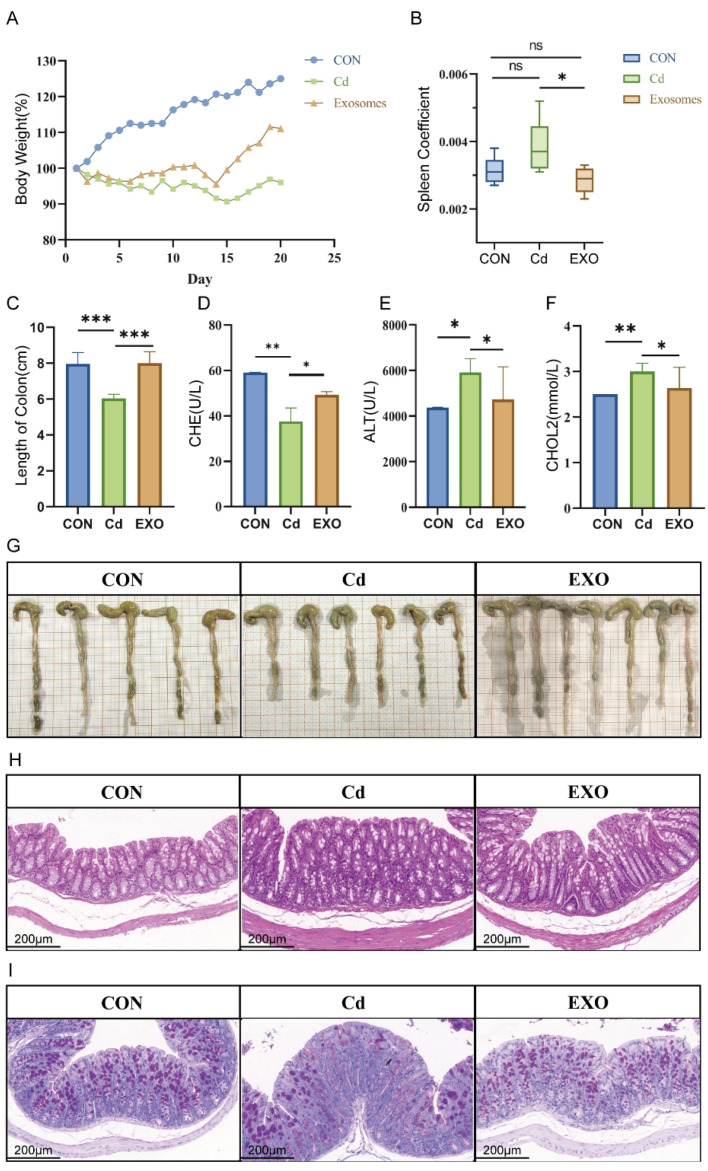
In vivo colitis treatment effect of SM-Exo through oral administration. (**A**) Daily body weight of FMT mice. Data are mean ± SD (*n* = 8). (**B**) Spleens of mice in CON, Cd, and Exo groups were weighted, and the spleen coefficient was calculated (spleen weight (mg)/body weight (g)). (**C**) Representative extracted colon length of FMT mice on day 20. Data are mean ± SD (*n* = 8). (**D**) Concentrations of serum CHE (U/L) among the CON, Cd, and EXO groups. Data are mean ± SD (*n* = 8). (**E**) Concentrations of serum ALT (U/L) among the CON, Cd, and EXO groups. Data are mean ± SD (*n* = 8). (**F**) Concentrations of serum CHOL2 (mmol/L) among the CON, Cd, and EXO groups. Data are mean ± SD (*n* = 8). (**G**) Representative extracted colon images of FMT mice on day 20. (**H**) Representative histopathological images of colon tissues stained with hematoxylin and eosin (H&E) on day 28. (**I**) AB-PAS staining of FMT mice colon. *** *p* < 0.001, ** *p* < 0.01, * *p* < 0.05, ns *p* > 0.05.

**Figure 3 ijms-26-03299-f003:**
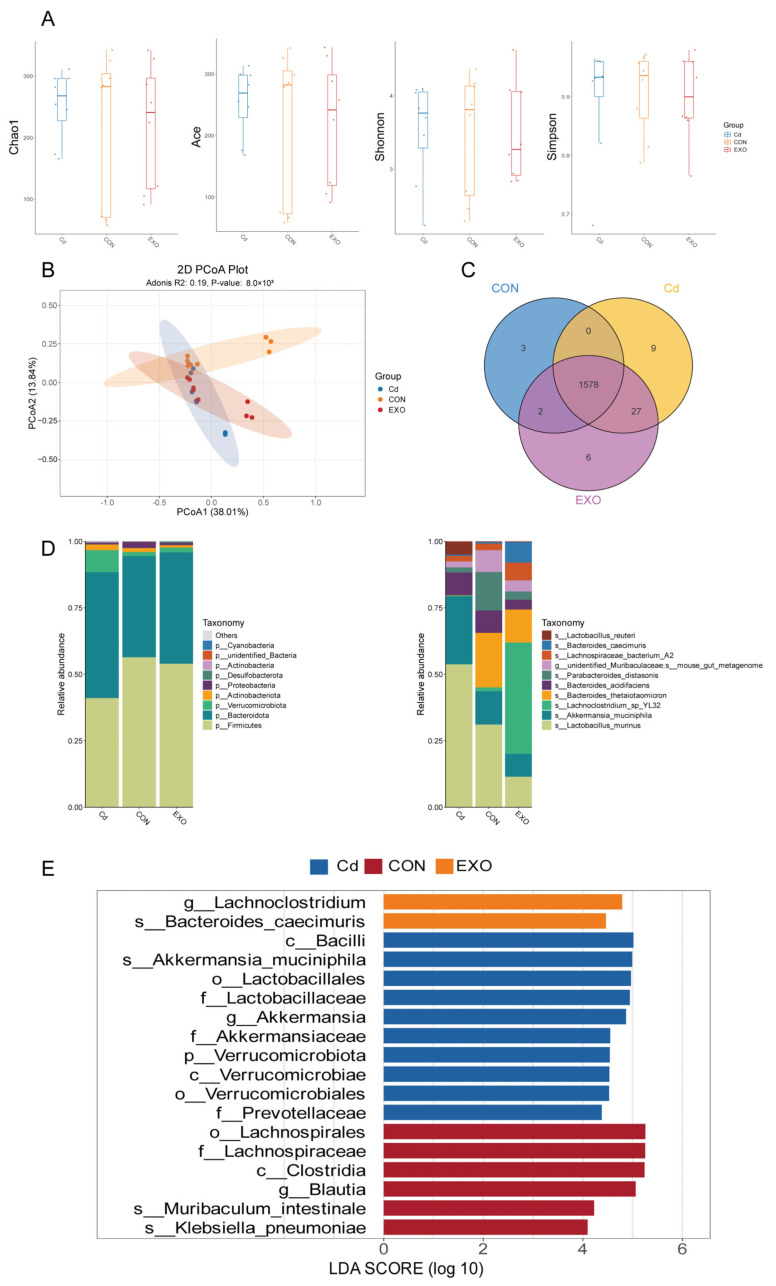
Microbial results of FMT mice colon content. (**A**) α diversity of CON, Cd, and EXO groups (*p* < 0.05). (**B**) β diversity of CON, Cd, and EXO groups (*p* < 0.05). (**C**) Venn diagram of microorganisms of CON, Cd, and EXO groups. (**D**) Microbial composition differences among the CON, Cd, and EXO groups at the phylum and genus levels. (**E**) Fold change diagram of differential microorganisms among CON, Cd, and EXO groups.

**Figure 4 ijms-26-03299-f004:**
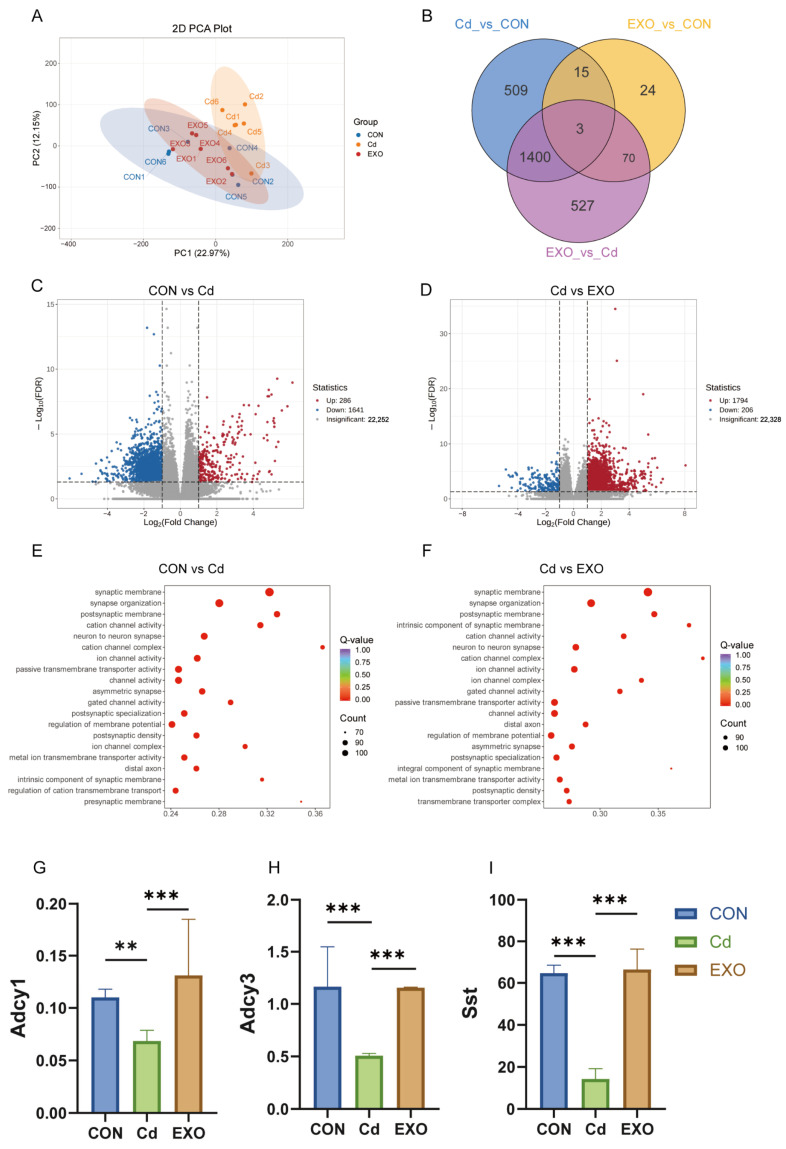
RNA-seq analysis was used to explore the mechanism underlying the SM-Exo treatment effect on cadmium-induced colitis in FMT mice. (**A**) PCA map of differentially expressed genes in CON, Cd, and EXO groups. (**B**) Venn diagram of differentially expressed genes in CON, Cd, and EXO groups. (**C**) Volcano plot of differentially expressed genes between CON and Cd groups. (**D**) Volcano plot of differentially expressed genes between Cd and EXO groups. (**E**) KEGG enrichment map of differentially expressed genes between CON and Cd groups. (**F**) KEGG enrichment map of differentially expressed genes between Cd and EXO groups. The data are presented as the mean ± SEM (*n* = 3 mice per group). The expression of Adcy1 (**G**), Adcy3 (**H**), Sst (**I**) *** *p* < 0.001, ** *p* < 0.01.

**Figure 5 ijms-26-03299-f005:**
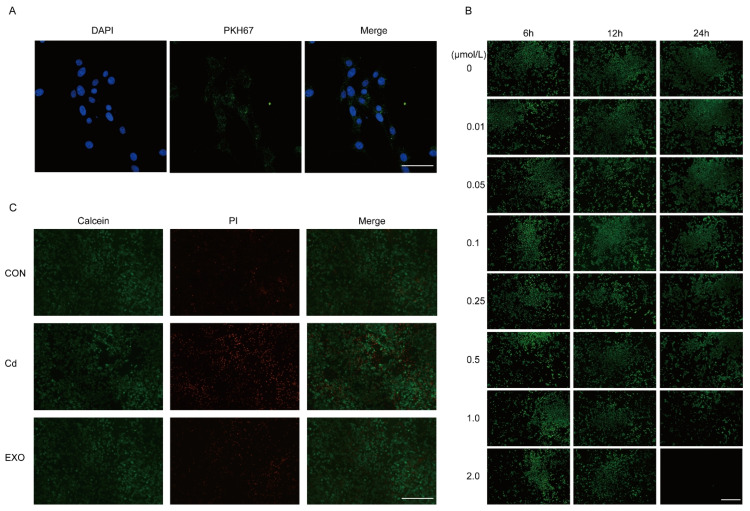
SM-Exo ameliorates CD-induced inflammation and death in cells. (**A**) The immunofluorescent staining of DAPI (blue immunofluorescence), PKH67 (green immunofluorescence). Bars, 50 µm. (**B**) The immunofluorescent staining Calcein AM (Green) of living MECE cells with concentration gradient of cadmium during 6 h, 12 h, and 24 h. Bars, 100 µm (**C**) The immunofluorescent staining Calcein AM of living MECE cells and PI (Red) of dead cells. Bars, 100 µm.

## Data Availability

Data are available upon reasonable request.

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
