# Peer review of "Oral Sheep Milk-Derived Exosome Therapeutics for *Cadmium*-Induced Inflammatory Bowel Disease"

_ijms, 2025, doi:10.3390/ijms26073299_

Round 1

Reviewer 1 Report

Comments and Suggestions for Authors

This study evaluates the therapeutic effects of sheep milk-derived exosomes (SM-Exo) in a mouse model of cadmium (Cd)-induced colitis, analyzing gut microbiota alterations, inflammatory markers, and gene expression changes. The authors propose that SM-Exo exert protective effects by modulating gut microbiota and activating the cAMP signaling pathway, suggesting a potential role in inflammatory bowel disease (IBD) management. The study is innovative and clinically relevant, given the growing interest in exosome-based therapies for gut inflammation. The methodology is generally sound, but several key areas require improvement, particularly regarding exosome characterization, mechanistic validation, and comparative analysis with existing therapies. I recommend major revisions before acceptance to strengthen the scientific rigor and translational impact of the study.

Major Concerns a

  • The Western blot analysis confirms the presence of exosomal markers (CD63, CD81, TSG101) and the absence of Calnexin, supporting the purity of the exosome preparation. However, there is no characterization of the exosomal cargo, which is crucial to understanding the underlying mechanism of action. Given that exosomes mediate biological effects through their RNA and protein content, it is important to analyze whether SM-Exo contain specific miRNAs, mRNAs, or proteins responsible for gut microbiota modulation and inflammation reduction. The authors should include RNA sequencing or proteomic profiling of SM-Exo to identify key bioactive molecules that may be responsible for the observed effects.
  • The authors claim that SM-Exo treatment increases cAMP-related gene expression (Adcy1, Adcy3, Sst), but functional validation is missing. Were intracellular cAMP levels directly measured? Gene expression alone is not sufficient to confirm pathway activation. Were downstream targets of cAMP signaling (e.g., CREB phosphorylation, PKA activation) analyzed? The authors should quantify intracellular cAMP levels or assess phosphorylation of key downstream effectors (e.g., CREB, PKA) to confirm pathway activation.
  • The microbiota analysis shows shifts in bacterial taxa (increase in Lachnoclostridium, changes in Bacteroidota and Verrucomicrobia), but no mechanistic connection is established between these changes and colitis improvement. Were bacterial-derived metabolites (e.g., short-chain fatty acids, bile acids) measured? These could provide a functional explanation for how microbiota restoration contributes to inflammation resolution. Did SM-Exo directly affect bacterial growth in vitro, or were these microbiota changes secondary to reduced inflammation? A metabolomic analysis of microbial-derived metabolites could help explain how gut microbiota shifts mediate the therapeutic effects of SM-Exo.
  • The manuscript suggests that SM-Exo could be a novel therapy for colitis, but does not compare its efficacy with standard treatments such as aminosalicylates, corticosteroids, or biologics. How does the reduction in inflammatory markers with SM-Exo compare to conventional IBD therapies? Would SM-Exo be used as a standalone treatment or as an adjunct to existing therapies? The authors should contextualize their findings within the current therapeutic landscape of IBD, discussing whether SM-Exo provide distinct advantages or limitations compared to standard treatments.
  • The study design appears well-structured, but there is no mention of sample size calculations to justify the number of animals used per group. Did the authors perform a power analysis to determine whether the study was adequately powered to detect significant effects? Were microbiome differences adjusted for multiple comparisons (e.g., Benjamini-Hochberg correction)? Given the complexity of microbiome data, controlling for false discovery rates is crucial. The authors should justify their sample size and include details on statistical adjustments for microbiome comparisons.

Reviewer 2 Report

Comments and Suggestions for Authors

In this manuscript, the authors report the effects of exosomes derived from sheep milk to treat cadmium-induced colitis in a murine experimental model, showing the changes in microbiota and gene expression. 

However, it is not clear the field of application of these findings, since the abstract is not specific, the introduction starts mentioning this is a problem of livestock, but finishes mentioning that could be useful for humans and other mammals.

Likewise, exosomes derived from sheep milk are referred in different ways, as SM-Exo in the abstract, M-exosomes and Sheep M-Exo in the Results section. Please unify. 

Please find below specific comments and suggestions, in the order they appear in the text.

In the abstract, please mention that Cd exposure leads to gastrointestinal diseases in livestock, or sheep, to avoid misleading the reader outside the field of application of the research.

It is not clear in the abstract whether the Cd was administered to the sheep from which the microbiota came for the FMT or the recipient mice.

In the introduction section, the information exposed that support exosomes for the treatment of intestinal diseases is taken out of context by not mentioning the organisms in which they have been proved. Likewise, they are mentioned as drug-delivery alternatives, while the focus of the study is in the role of exosomes per se in Cd induced colitis.

At the beginning of the second paragraph, it is not complete the first sentence "Oral administration..." since it does not refer to any treatment or medication.

Likewise, when referring to the goat milk-derived exosomes, please mention that they were shown effective in mice.

It is not clear in this paragraph to what "oral drug delivery" you are referring, no drugs are mentioned.

In the results section, figure 1 shows the western blots fro CD63, CD 81, TSG101 and Calnexin, however, the text mentions CD9, but not TSG101.

In point 2.2, it is first referred to the differences in % of weight gain between the trhee groups of mice, and then after 14 days, but no data is mentioned. Were the 14 days after the first 20 days?

Related to figure 3, panel A, please mention the p value of the significant differences mentioned in the text, either on the text or the figure. Related to panel C of the same figure, it is not clear what the numbers in each circle are representing, as "unique microorganismos", as mentioned in the text, is ambiguos, are you referring to OTUs or ASVs? Panel D mentions "Changes in microbial abundance of CON, Cd and EXO group at the phylum and genus levels",  which implies a comparison between before and after intervention, however, the panel just shows differences at the phylum and genus level between groups.

On figure 5, please add scale bars to the photographs. Likewise, please explain how you concluded that CD induced inflammation from the pictures depicted in this figure.

The methods followed to achieve the results of figure 5 are not detailed in the method section.

In the discussion section, although the authors show that SM-Exo treatment benefits the gut microbiota in the murine model, please argue how the results support "that SM-Exo administration restores a healthy microbial profile", considering that Figure 3D shows remarkable differences in the relative abundances at the genus level.

Please discuss what is in the exosomes that can cause the effects observed, is it just the presence or the proteins and other biomolecules inside them or on their surface? 

In the conclusion, please provide more details on how your data can highlight "the complex interaction between host genetics, immune responses, and the gut microbiome in cadmium-induced colitis". The data presented did not analyze host genetics nor the immune responses. Likewise, the limitations of the study and the field of application is not clearly mentioned in the discussion. For instance, the study was performed on a murine model of colitis induced through the FMT from sheep exposed to cadmium. The murine model was not germ-free, but treated with antibiotics, which do not assure complete elimination of host microbiota. Furthermore, if trying to broaden the possible application of these results to human diseases like IBD, the diet of sheep, mouse, and humans is very different, so the microbiota and probably the biomolecule's content of milk exosomes in each species.

Considering that mice received FMT along with the exosomes, can this be a factor that could have influenced the result? Can exosomes could be targeting the microbiota before it was stablished? Could administration of the milk exosomes days after or before FMT produce different results? This considering that the exosomes could function as a treatment when the colitis is already stablished, or as a profilacting agent to prevent colitis.

In the methodology section, there are missing details of the methodology followed in multiple points, please revise.

For example, in methodology point 4.2, please add the number of reference por "Feng et al.".

Western blots (point 4.8) mention that total protein from colon tissues was used; however, the results section (point 2.1) implies that the western blots were performed with the exosome proteins. Please revise. Likewise, please include the total quantity of protein added to the SDS-PAGE gel, the dilution of the primary antibodies, and the secondary andtibodies used and their dilution.

In point 4.9, there are no details on the construction of cDNA libaries.

Point 4.1016. is probably 4.10. 

It is mentioned that a metabolomic analysis was performed on the discussion and methods, but there are no results from it.

As for the supporting information (blot images), what sample is in each line, and how did you determine the molecular weight of the bands from those images, since no molecular weight ladder is visible? Likewise, the images are cut just to show a single band. Were there any other bands in each lane? Was any positive or negative control included?

Round 2

Reviewer 1 Report

Comments and Suggestions for Authors

The authors have adequately responded to all comments and the manuscript can now be considered for publication. 

Reviewer 2 Report

Comments and Suggestions for Authors

The author's replied satisfactorily to the provided suggestions and concerns.

Just a minor comment: Please homogenize "Cd" or"CD". For example, in lines 132 and 145, "Cd" is used, while in lines 130 and 134, "CD" is used. Likewise, some figures and figure legends include either Cd or CD.